# PELEE: A REAL-TIME OBJECT DETECTION SYSTEM ON MOBILE DEVICES

**Robert J. Wang, Xiang Li, Shuang Ao & Charles X. Ling**
Department of Computer Science
University of Western Ontario
London, Ontario, Canada, N6A 3K7
{jwan563,lxiang2,sao,charles.ling}@uwo.ca

## ABSTRACT

An increasing need of running Convolutional Neural Network (CNN) models on mobile devices with limited computing power and memory resource encourages studies on efficient model design. A number of efficient architectures have been proposed in recent years, for example, MobileNet, ShuffleNet, and NASNet-A. However, all these models are heavily dependent on depthwise separable convolution which lacks efficient implementation in most deep learning frameworks. In this study, we propose an efficient architecture named PeleeNet, which is built with conventional convolution instead. On ImageNet ILSVRC 2012 dataset, our proposed PeleeNet achieves a higher accuracy by 0.6% (71.3% vs. 70.7%) and 11% lower computational cost than MobileNet, the state-of-the-art efficient architecture. Meanwhile, PeleeNet is only half of the model size of MobileNet. We then propose a real-time object detection system by combining PeleeNet with Single Shot MultiBox Detector (SSD) method and optimizing the architecture for fast speed. Our proposed detection system[1], named Pelee, achieves 70.9% mAP (mean average precision) on PASCAL VOC2007 dataset at the speed of 17.1 FPS on iPhone 6s and 23.6 FPS on iPhone 8. Compared to TinyYOLOv2, our proposed Pelee is more accurate (70.9% vs. 57.1%), 1.88 times lower in computational cost and 1.92 times smaller in model size.

## 1 INTRODUCTION

There has been a rising interest in running high-quality CNN models under strict constraints on memory and computational budget. Many innovative architectures, such as MobileNets (Howard et al. (2017)), ShuffleNet (Zhang et al. (2017)), NASNet-A (Zoph et al. (2017)), have been proposed in recent years. However, all these architectures are heavily dependent on depthwise separable convolution (Szegedy et al. (2015)) which lacks efficient implementation. Meanwhile, there are few studies that combine efficient models with fast object detection algorithms. This research tries to explore the design of an efficient CNN architecture for both image classification tasks and object detection tasks. It has made a number of major contributions listed as follows:

**We propose a variant of DenseNet (Huang et al. (2016a)) architecture called PeleeNet for mobile devices.** PeleeNet follows the innovate connectivity pattern and some of key design principals of DenseNet. It is also designed to meet strict constraints on memory and computational budget. Experimental results on Stanford Dogs (Khosla et al. (2011)) dataset show that our proposed PeleeNet is higher in accuracy than the one built with the original DenseNet architecture by 5.05% and higher than MobileNet (Howard et al. (2017)) by 6.53%. PeleeNet achieves a compelling result on ImageNet ILSVRC 2012 (Deng et al. (2009)) as well. The top-1 accuracy of PeleeNet is 71.3% which is higher than that of MobileNet by 0.6%. It is also important to point out that PeleeNet is only half of the model size of MobileNet. Some of the key features of PeleeNet are:

- **Two-Way Dense Layer** As shown on Fig. 1.a, we use a 2-way dense layer to get different scales of receptive fields. One way of the layer uses a small kernel size (3x3), which is

---

[1]The code and models are available at: https://github.com/Robert-JunWang/Pelee

good enough to capture small-size objects. The other way of the layer uses two stacked 3x3 convolution to learn visual patterns for large objects.

- **Stem Block** Motivated by Inception-v4 (Szegedy et al. (2017)) and DSOD (Shen et al. (2017)), we design an cost efficient stem block before the first dense layer. The structure of stem block is shown on Fig. 1.b. This stem block can effectively improve the feature expression ability without adding computational cost too much - better than other more expensive methods, e.g., increasing channels of the first convolution layer or increasing growth rate.

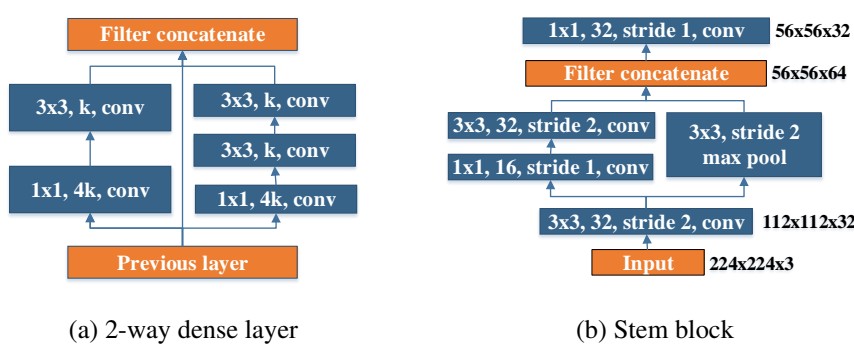

(a) 2-way dense layer      (b) Stem block

Figure 1: Structure of 2-way dense layer and stem block

- **Dynamic Number of Channels in Bottleneck Layer** Another highlight is that the number of channels in the bottleneck layer varies according to the input shape to make sure the number of output channels does not exceed the number of its input channels. Compared to the original DenseNet structure, our experiments show that this method can save up to 28.5% of the computational cost with a small impact on accuracy.

- **Transition Layer without Compression** Our experiments show that the compression factor proposed by DenseNet hurts the feature expression. We always keep the number of output channels the same as the number of input channels in transition layers.

- **Composite Function** To improve actual speed, we use the conventional wisdom of post-activation (Convolution - Batch Normalization (Ioffe & Szegedy (2015)) - Relu) as our composite function instead of pre-activation used in DenseNet. To compensate for the negative impact on accuracy caused by this change, we use a shallow and wide network structure. We also add a 1x1 convolution layer after the last dense block to get the stronger representational abilities.

**We optimize the network architecture of Single Shot MultiBox Detector (SSD) (Liu et al. (2016)) for speed acceleration and then combine it with PeleeNet.** Our proposed system, named Pelee, achieves 70.9% mAP on PASCAL VOC (Everingham et al. (2010)) 2007 object detection dataset. It also outperforms Tiny-YOLOv2 (Redmon & Farhadi (2016)), the most widely used computational efficient object detection system, in terms of accuracy, speed and model size. The major enhancements proposed to balance speed and accuracy are:

- **Feature Map Selection** We build object detection network in a way different from the original SSD with a carefully selected set of 5 scale feature maps (19 x 19, 10 x 10, 5 x 5, 3 x 3, and 1 x 1). To reduce computational cost, we do not use 38 x 38 feature map.

- **Residual Prediction Block** We follow the design ideas proposed by Lee et al. (2017) that encourage features to be passed along the feature extraction network. For each feature map used for detection, we build a residual (He et al. (2016)) block before conducting prediction.

- **Small Convolutional Kernel for Prediction** Residual prediction block makes it possible for us to apply 1x1 convolutional kernels to predict category scores and box offsets. Our experiments show that the accuracy of the model using 1x1 kernels is almost the same as

that of the model using 3x3 kernels. However, 1x1 kernels reduce the computational cost by 21.5%.

**We provide an efficient implementation of SSD algorithm on iOS.** We have successfully ported SSD to iOS and provided an optimized code implementation. Our proposed system runs at the speed of 17.1 FPS on iPhone 6s and 23.6 FPS on iPhone 8. The speed on iPhone 6s, a phone released in 2015, is 1.6 times faster than that of the official SSD implementation on a server with a powerful Intel i7-6700K@4.00GHz CPU.

## 2 EXPERIMENTAL RESULTS

### 2.1 RESULTS ON STANFORD DOGS AND IMAGENET ILSVRC 2012

Our PeleeNet is trained by PyTorch with mini-batch size 256 for 120 epochs. The model is trained with a cosine learning rate annealing schedule which starts from 0.1, similar to what is used by Pleiss et al. (2017) and Loshchilov & Hutter (2016).

Table 1: Top-1 accuracy on Stanford Dogs and ILSVRC 2012. This Stanford Dogs dataset is a subset of ILSVRC 2012 according to the ImageNet wnid used in the original Stanford Dogs.

| Model | Million MACs | Million Parameters | Stanford Dogs | ILSVRC 2012 |
|---|---|---|---|---|
| **1.0 MobileNet** | 569 | 4.24 | 73.5 | 70.7 |
| **ShuffleNet 2x (g = 3)** | 524 | - | - | 70.9 |
| **NASNet-A** | 564 | 5.3 | - | 74.0 |
| **DenseNet-41** | 543 | 1.07 | 75.02 | - |
| **PeleeNet** (ours) | 507 | 2.09 | **80.03** | **71.3** |

### 2.2 RESULTS ON VOC 2007

Our object detection system is based on the source code of SSD[2] and is trained with Caffe (Jia et al. (2014)). As can be seen from Table 2, the accuracy of Pelee is higher than that of TinyYOLOv2 by 13.8% and higher than that of SSD+MobileNet (Huang et al. (2016b)) by 2.9%. It is even higher than that of YOLOv2-288 at only 15.2% of the computational cost of YOLOv2-288.

Table 2: Results on PASCAL VOC 2007

| Model | Million MACs | Million Parameters | mAP | Speed (FPS) | | |
|---|---|---|---|---|---|---|
| | | | | iPhone6s | iPhone8 | Intel i7 |
| **Tiny-YOLOv2** | 3490 | 15.86 | 57.1 | 9.3 | 23.8 | 2.4 |
| **YOLOv2-288** | 7940 | 57.96 | 69.0 | - | - | 1.0 |
| **SSD+MobileNet** | 1150 | 5.77 | 68 | 16.1 | 22.8 | 6.1 |
| **Pelee** (ours) | 1210 | 5.43 | **70.9** | **17.1** | **23.6** | **6.7** |

## 3 CONCLUSION

Depthwise separable convolution is not the only way to build an efficient model. Instead of using depthwise separable convolution, our proposed PeleeNet and Pelee are built with conventional convolution and have achieved compelling results on ILSVRC 2012 and VOC 2007.

By combining efficient architecture design with mobile GPU and hardware-specified optimized run-time libraries, we are able to perform real-time prediction for image classification and object detection tasks on mobile devices. For example, Pelee, our proposed object detection system, can run 17.1 FPS on iPhone 6s and 23.6 FPS on iPhone 8 with high accuracy.

---

[2]https://github.com/weiliu89/caffe/tree/ssd

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
