# OpenReview forum: "Pelee: A Real-Time Object Detection System on Mobile Devices"
_ICLR.cc/2018/Workshop — Accept_

### Official Review · AnonReviewer1 · 2018-03-10
**-**

**Rating:** 6
**Confidence:** 4

**Review:**

This paper proposes a new convnet architecture intended to be accurate yet efficient for inference on mobile devices. It does so using computationally efficient convolutional blocks involving bottleneck layers. While somewhat incremental, the results in terms of accuracy (for classification and detection) and computational efficiency compare favorably with other models designed for efficient mobile inference*, and good performance on mobile devices is a topic of wide interest.  The code and models are available. A few comments:

- The bottleneck structure of the 2-way dense layer is similar to that of ResNets [He et al., 2016]. ResNets should be cited as related work in the description of the module at the least.  Ideally there would also be a direct comparison to the ResNet module design vs. the proposed 2-way dense layer in terms of computational efficiency and accuracy, but I don’t think this is necessary for a workshop submission.

- I wasn’t able to understand the “Residual Prediction Block” description -- it should be clarified.  E.g., how many residual blocks are there (just 1?)? What are its inputs? Why not use residuals elsewhere?

- What are “MACs” (as used in the results Tables 1 and 2)?  This should be defined.

(* I have not closely followed the progress on efficient mobile convnet architectures and am assuming the alternatives this work compares against -- MobileNet, DenseNet, TinyYOLO, etc. -- are the state of the art.)

---

### Official Review · AnonReviewer3 · 2018-03-12
**Nice work on improving computing constraint object-detection architectures**

**Rating:** 7
**Confidence:** 4

**Review:**

This work addresses the task of creating high quality object detection system at low computational cost.

The heart of this work is a new modular convolutional block (similar to Inception) that is designed for more efficient processing of computer vision signals, especially for (but not limited to) object detection.

The authors report significant improvements over MobileNet based SSD at a similar or lower computational cost.

This work is original and the paper is clearly and well written, clearly suitable for a workshop submission.

---

### Decision · Program_Chairs · 2018-03-20
**ICLR 2018 Workshop Acceptance Decision**

**Decision:**

Accept

**Comment:**

Congratulations, your paper was accepted to the ICLR workshop.